# Systematic literature review reveals suboptimal use of chemical probes in cell-based biomedical research

Jayden Sterling[1], Jennifer R. Baker[2], Adam McCluskey [2] & Lenka Munoz [1] ✉

Chemical probes have reached a prominent role in biomedical research, but their impact is governed by experimental design. To gain insight into the use of chemical probes, we conducted a systematic review of 662 publications, understood here as primary research articles, employing eight different chemical probes in cell-based research. We summarised (i) concentration(s) at which chemical probes were used in cell-based assays, (ii) inclusion of structurally matched target-inactive control compounds and (iii) orthogonal chemical probes. Here, we show that only 4% of analysed eligible publications used chemical probes within the recommended concentration range and included inactive compounds as well as orthogonal chemical probes. These findings indicate that the best practice with chemical probes is yet to be implemented in biomedical research. To achieve this, we propose 'the rule of two': At least two chemical probes (either orthogonal target-engaging probes, and/or a pair of a chemical probe and matched target-inactive compound) to be employed at recommended concentrations in every study.

Chemical probes are well-characterised small molecules with potency and selectivity for a protein of interest[1,2]. The term 'chemical probe' distinguishes compounds used in basic and preclinical research from 'drugs' used in the clinic, from the terms 'inhibitor', 'ligand', 'agonist' or 'antagonist' which are molecules targeting a given protein but are insufficiently characterised, and also from the term '*probes*' which is often referring to laboratory reagents for biophysical and imaging studies[3]. Chemical probes are tools to understand the function of a targeted protein at the mechanistic level and have also reached a prominent role in target validation research. The development of first-in-class drugs often begins with the identification of a new target from -omics screens. In the target validation stage, various strategies are employed to evaluate whether the identified target has a key role in the disease process and whether its pharmacological modulation could lead to a therapeutic efficacy. In this stage, chemical probes are complementary to molecular probes such as CRISPR and RNA interference, because they offer unique advantages. Unlike molecular probes which induce target's knock-down or knock-out over a longer timeframe, chemical probes rapidly inhibit the activity of a protein of interest.

Furthermore, when coupled with molecular probes, chemical probes can distinguish between effects due to the target's presence in the cell, and effects due to the inhibition of catalytic or protein-protein interaction activity[4].

The necessity of chemical probes for mechanistic and target validation research[1–10] has led to a steadily growing number of high-quality chemical probes[10]. Every chemical probe must satisfy the minimal fundamental criteria, known as fitness factors: namely potency, selectivity, and cellular activity[1,9]. While these properties may vary based on the nature of the targeted protein; in principle, chemical probes adhere to the in vitro potency of less than 100 nM, selectivity for the targeted protein being at least 30-fold against sequence-related proteins of the same family, and on-target cellular activity at concentrations ideally below 1 µM[1,2,9,11]. Similarly, the guidelines for the use of chemical probes in cell-based experiments may vary due to the different nature of the hypotheses being investigated. Nevertheless, the foremost recommendation is to use chemical probes at concentrations closest to the validated on-target effect[1,9]. Even the most selective chemical probe will become non-selective if used at a high concentration. Second, it is

[1]Faculty of Medicine and Health, Charles Perkins Centre, The University of Sydney, Camperdown, NSW 2006, Australia. [2]Discipline of Chemistry, School of Environmental and Life Sciences, The University of Newcastle, Callaghan, NSW 2308, Australia. ✉e-mail: lenka.munoz@sydney.edu.au

strongly encouraged that the chemical probe be accompanied by a target-inactive, but structurally similar analogue to serve as a negative control. Third, each protein should be targeted by another well-characterized orthogonal chemical probe having a different chemical structure. Thus, for any given protein, the ideal cell-based experiment would employ at least two orthogonal chemical probes at reasonable concentrations and be accompanied, when available, by structurally related but target-inactive derivatives[8,9,11,12].

There are many factors to consider when selecting and using a chemical probe to address a hypothesis, which can be challenging, especially for non-experts. To facilitate the correct use of chemical probes in biomedical research, open-access resources have been established by the chemical biology community[11]. The Chemical Probes Portal (www.chemicalprobes.org)[9], a user-friendly platform for non-chemists, at the time of writing includes 547 chemical probes that cover over 400 protein targets[13]. A total of 321 chemical probes have three or more stars and are therefore specifically recommended by the Portal to confidently study 281 protein targets. Furthermore, the Portal also lists an additional 248 compounds labelled as 'Historical Compounds'. These compounds should not be used to study the function of specific proteins as they are either seriously flawed or are outdated and superseded. This important resource is complemented by the Chemical Probes website of the Structural Genomics Consortium (https://www.thesgc.org/chemical-probes) and the Donated Chemical Probes website (www.sgc-ffm.uni-frankfurt.de)[14], where a consortium of pharmaceutical companies offer access to their previously undisclosed chemical probes. While these portals are based on peer-review and recommendations by chemical biology experts, the Probe Miner (https://probeminer.icr.ac.uk/), a database of over 1.8 million small molecules, comprehensively analyses peer-reviewed literature and provides a relative ranking of the chemical probe based on objective statistical assessment of the large scale data[15]. Thus, the assessment of a chemical probe provided by the Probe Miner is a valuable complement to the experts' review basis of the above mentioned portals. Finally, the Probes & Drugs database (www.probes-drugs.org)[16] contains over 4600 probes, of which over 1100 probes are approved by the Probes & Drugs community. In summary, these online resources provide information about chemical probes and some also list recommendations such as maximal in-cell concentration, availability of matched target-inactive control molecule, and orthogonal chemical probes. Adherence to these recommendations is crucial to generate robust findings.

The problem of inaccurate data has been publicly noted more than a decade ago. While reasons are diverse[17,18], the suboptimal use of chemical probes has emerged as one of the culprits in the robustness crisis[3]. However, while the suboptimal use of chemical probes has been widely discussed, the extent of this problem remains unknown. Therefore, we analysed how selected chemical probes targeting epigenetic and kinase targets - histone methyltransferases EZH2 and G9a/

GLP, histone demethylase KDM6, histone acetyltransferase CBBP/p300 and kinases Aurora, mTOR and CDK7 - have been applied in primary research articles (referred to as publications herein). In our analysis, we considered three questions: i) Has the chemical probe been used in cellular assays within the recommended concentration range? ii) If a matched target-inactive control molecule is available, has it been employed in cellular assays? iii) If orthogonal inhibitors targeting the protein of interest exist, have they been employed in cellular assays? Here, we show that while the frequency of chemical probes in research is encouragingly high, a worryingly low fraction of the analysed publications used chemical probes correctly.

## Results
### Study selection
Eight chemical probes (Table 1) were selected because they target proteins representing research fields of different sizes. PubMed search (Dec 2022) using the 'Advanced Search' function to include all synonyms of the chemical probes' primary targets retrieved 5908 (EZH2); 21,977 (G9a/GLP); 1656 (KDM6A); 20,733 (CREBBP); 54,733 (Aurora); 45,728 (mTOR) and 768 (CDK7) articles. While we provide a brief overview of all chemical probes for a given target, in the selection of chemical probes for the systematic review analysis, preferences were given to (i) 'older' probes disclosed at least five years ago; (ii) probes with matched target-inactive control compounds and/or (iii) commercially available chemical probes. Another factor was to select chemical probes with a manageable number of records. For example, PubMed search (Dec 2022) for the keyword 'rapamycin' (a chemical probe targeting mTOR kinases) retrieved over 51,000 records; it is not feasible to manually review such literature corpora. All selected chemical probes received three- or four-star recommendations in the Chemical Probes Portal and thus are endorsed by the Scientific Expert Review Panel to study the function of their primary targets (Table 1). The Global Score in the Probe Miner places UNC0638, GSK-J4, AMG900, AZD1152, AZD2014 in the top 10%, whereas A-485 is placed in the top 12%; making these chemical probes one of the best tools to interrogate their primary targets. The Global Scores for UNC1999 and THZ1 are low, 0.15 and 0.29, respectively (Table 1).

We screened 1131 articles and identified 662 primary research articles (i.e., publications) eligible for the systematic review analysis (Fig. 1)[19]. All eligible publications contained at least one figure panel presenting results obtained with a given chemical probe used in a cell-based assay, irrespective of the topics and research fields. To address the compliance with the recommended experimental design for chemical probes[9,11], full-text contents across these 662 publications were analysed for chemical probes' concentration in cell-based assays, use of negative control compounds and orthogonal inhibitors (Fig. 1). Analysed publications were considered complying or partially complying if they employed a given chemical probe within the recommended in-cell concentration range, used matched target-inactive

## Table 1 | Overview of chemical probes selected for the study

| Probe | Primary target | Year of disclosure | Inactive control | Orthogonal chemical probes | Chemical Probes Portal score (Aug 2022) | Probe Miner global score & rank (Aug 2022) |
|---|---|---|---|---|---|---|
| UNC1999 | EZH2 | 2013 | UNC2400 | Supplementary Fig. 1 | 3 stars | 0.15 195/211 (top 93%) |
| UNC0638 | G9a/ GLP | 2011 | UNC0737 | Supplementary Fig. 3a | 3 stars | 0.54 19/329 (top 6%) |
| GSK-J4 | KDM6 | 2012 | GSK-J5 | Not available | 3 stars | 0.53 1/52 (top 2%)[a] |
| A-485 | CREBBP/p300 | 2017 | A-486 | Supplementary Fig. 6 | 3 stars | 0.39 35/302 (top 12%) |
| AMG900 | Aurora kinases | 2010 | Not available | Supplementary Fig. 8a | 4 stars | 0.58 20/2906 (top 1%) |
| AZD1152 | Aurora kinases | 2007 | Not available | Supplementary Fig. 8a | 4 stars | 0.75 1/1888 (top 1%) |
| AZD2014 | mTOR | 2013 | Not available | Supplementary Fig. 8b | 4 stars | 0.49 281/4230 (top 7%) |
| THZ1 | CDK7 CDK12/13 | 2014 | THZ-R1 | Supplementary Fig. 12 | 3 stars | 0.29 171/374 (top 46%) |

[a]The score and ranking refer to compound URY (https://probeminer.icr.ac.uk/#/O15054), which is the active hydrolysed GSK-J4 analogue structurally identical to GSK-J1.

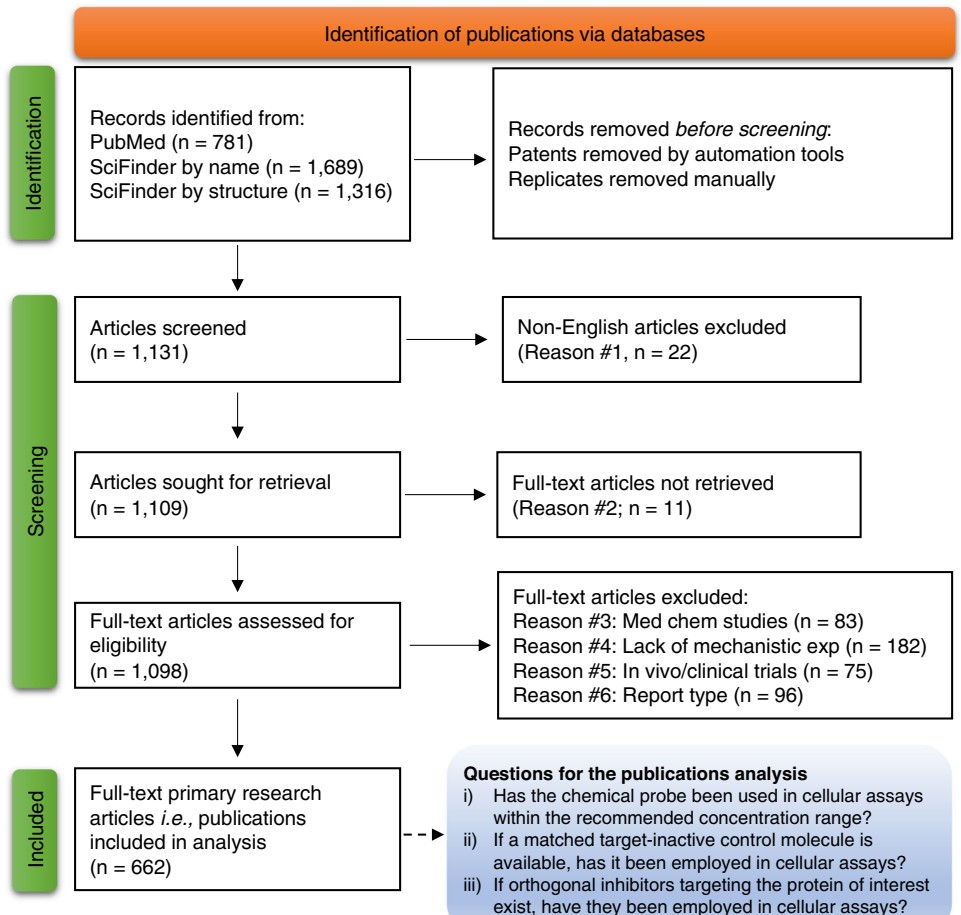

**Fig. 1 | Preferred items for systematic reviews and meta-analysis (PRISMA) flow diagram.** From 1131 records related to 8 chemical probes, we excluded articles (1) not written in English, (2) whose full text were not available via University of Sydney access; (3) describing the discovery of a given chemical probe and/or medicinal chemistry studies; (4) in which the chemical probe was used to generate chemical probe-resistant clones and/or articles lacking mechanistic experiments; (5) containing only in vivo data and/or clinical trial results; and 6) reviews, commentaries, editorials, letters to the editors, conference proceedings, pre-prints. We analysed 662 eligible publications for the concentration of chemical probes in cell-based assays, the inclusion of negative control compounds and the inclusion of orthogonal chemical probes. PRISMA flow diagrams for each individual chemical probe are provided in the Supplementary Information file.

control compound (where available), and at least one orthogonal inhibitor (where available) (Fig. 2). Details of publications analyses are provided in the Supplementary Tables 1–16 and presented for each chemical probe in subsections below.

### EZH2 chemical probe UNC1999

Methylation of lysine residues in histone tails by histone methyltransferases (KMTs) is a prevalent epigenetic modification. Enhancer of Zeste Homologue 2 (EZH2, also known as KMT6A) is the main catalytic subunit of the Polycomb Repressive Complex 2 responsible for mono-, di- and trimethylation of histone 3 (H3) at lysine (K) 27 (H3K27). EZH2 catalysed H3K27 methylation silences the transcription of nearby genes. EZH2 is often replaced by its closely related homologue EZH1 (also known as KMT6B) in differentiated and quiescent cells. EZH1 and EZH2 share 63% overall homology and 94% of their catalytic SET domain; hence EZH2 targeting chemical probes also target EZH1 with some potency[20,21].

An early tool compound targeting EZH2 was EPZ005687[22], which while a potent EZH2 inhibitor, lacked the minimal fundamental criteria of a chemical probe. The first published EZH2 inhibitor now endorsed as a chemical probe is EI1 (Supplementary Fig. 1)[23]. Optimisation of the EPZ005687's pharmacophore yielded chemical probes GSK343[24], EPZ-6438[25] (and the structurally-related EPZ011989[26]) as well as UNC1999[27]. UNC1999 is particularly noteworthy as it is one of the only two

available EZH2 targeting chemical probes to also have a dedicated inactive matched partner[27]. UNC2400 has an *N*-methylated benzamide core to distinguish itself from its active counterpart UNC1999. This *N*-methylation makes UNC2400 over 1000-fold less potent than UNC1999, rendering it as a negative control in cell-based experiments. The indazole-based JQEZ5 was designed as an open-source chemical probe[28] and is another chemical probe to also have a structurally matched inactive analogue: JQEZ23 contains a methyl-substituted pyridine in place of JQEZ5's pyridino-moiety (Supplementary Fig. 1). CPI-169 and CPI-360[29] contain a 2-methyl indole core and differ only with the alkylation substitution of the indole amine. The third chemical probe in this family CPI-1205[30] also features the 2-methyl indole common to both CPI-169 and CPI-360 (Supplementary Fig. 1).

We conducted a systematic review of 49 eligible publications (Supplementary Fig. 2) presenting at least one figure panel using UNC1999 in a cell-based assay (Supplementary Table 2). As the recommended maximal in-cell concentration for UNC1999 varies between the Chemical Probes Portal and the Structural Genomics Consortium sites (400 nM and 3 µM, respectively), we analysed compliance with both concentrations. We found 5 publications (10%) employing UNC1999 below the 400 nM concentration across most experiments. When considering the 3 µM recommendation, in total 18 publications (37%) fully complied; and 14 publications (29%) were considered as partially complying because they reported UNC1999 at

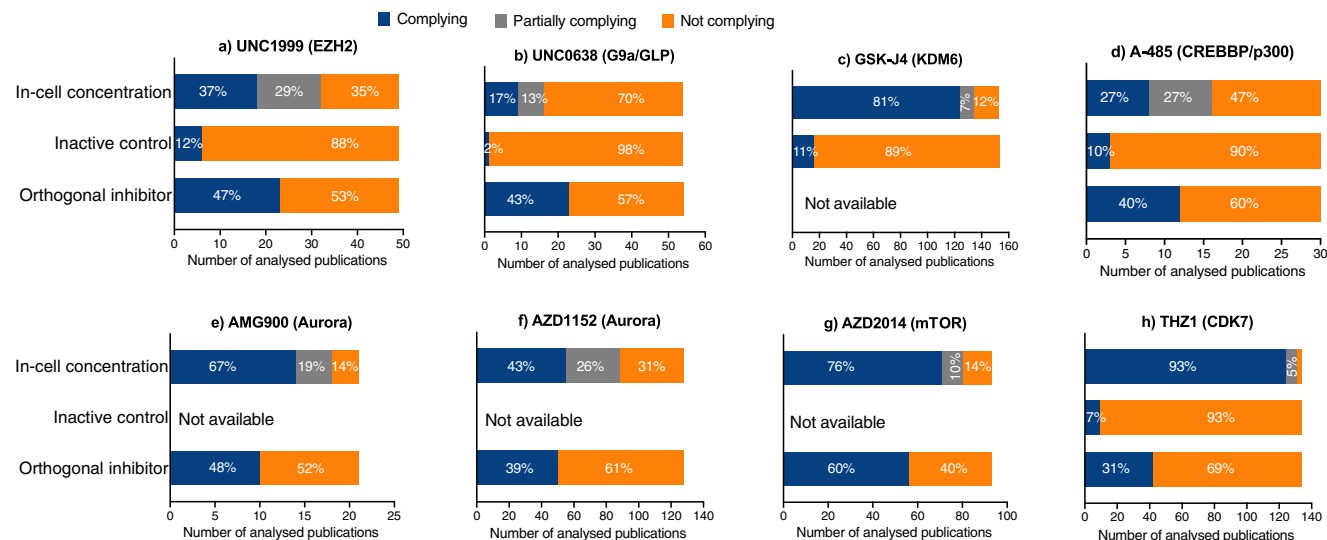

**Fig. 2 | Percentages of publications complying (blue), partially complying (grey) and not complying (orange) with guidelines for a given chemical probe.** The complying and partially complying publications employed a given chemical probe within the recommended in-cell concentration range: (**a**) UNC1999: up to 3 μM; (**b**) UNC0638: up to 250 nM; (**c**) GSK-J4: up to 5 &10 μM; (**d**) A-485: up to 1 μM; (**e**) AMG900: up to 100 nM; (**f**) AZD1152: up to 100 nM; (**g**) AZD2014: up to 2 μM; (**h**) THZ1: up to 1 μM), used matched target-inactive control compound (where available) and at least one orthogonal inhibitor (where available). [Note: The in-cell concentration percentages for UNC1999 and A-485 total to 101% due to rounding up individual percentages.] Source data are provided as a Source Data file.

concentrations below and above 3 μM. Of the analysed 49 publications, 6 publications (12%) included the inactive analogue UNC2400, and 23 publications (47%) performed experiments with orthogonal EZH2 inhibitors (Fig. 2a), some of which were endorsed chemical probes (e.g., GSK-343; Supplementary Table 2).

## G9a/GLP chemical probe UNC0638

G9a (also known as EHMT2 or KMT1C) and the closely related G9a-like protein (GLP, also known as EHMT1 or KMT1D) are methyltransferases responsible for mono- and di-methylation of H3K9, with H3K9me1/me2 marks silencing gene transcription. Compound BIX01294 is one of the first G9a/GLP inhibitors; however, because of its lack of selectivity and high cellular toxicity[31], there is a tight window between achieving cellular G9a/GLP target engagement while avoiding off-target toxicity. Follow-up chemistry based on the BIX01294 scaffold yielded G9a/GLP chemical probe UNC0638 (Supplementary Fig. 3a)[32]. The *N*-methyl analogue of UNC0638, compound UNC0737, is >300-fold less potent than UNC0638 and serves as a valuable inactive control[32]. Optimisation of UNC0638's short half-life led to the development of the first G9a/GLP in vivo chemical probe UNC0642[33]. Another chemical probe targeting G9a/GLP is A-366[34], which has a scaffold different from UNC0638 and UNC0642; hence A-366 provides a valuable orthogonal tool (Supplementary Fig. 3a).

To gain an insight into the G9a/GLP research, we reviewed 54 eligible publications (Supplementary Fig. 4) with at least one figure panel presenting the results of cell-based experiments performed with UNC0638 (Supplementary Table 4). We identified 9 publications (17%) that treated cells with UNC0638 within the recommended concentration range (up to 250 nM)[32] and 7 partially complying publications (13%) which employed UNC0638 at a higher concentration in some experiments. Of all analysed publications, only one publications (2%) included UNC0737 as a control compound and 23 (43%) publications used another G9a/GLP inhibitor alongside UNC0638 (Fig. 2b).

## KDM6 chemical probe GSK-J4

Histone lysine demethylase A (KDM6A, also known as UTX) and KDM6B (also known as JMJD3) remove H3K27 methylation marks, thereby activating gene transcription[35]. The liganded crystal structure of KDM6B enabled the development of GSK-J1, the first small-molecule

inhibitor of a histone lysine demethylase[36]. While the GSK-J1's acid moiety is critical for binding to KDM6B, it adversely affects cellular permeability. As such, the non-cell permeable GSK-J1 (listed as a Historic Compound; https://www.chemicalprobes.org/gsk-j1) should not be used in cell-based assays that depend on inhibition of KDM6A/B. Masking the polarity of the acid group with an ethyl ester moiety yielded chemical probe GSK-J4 (Supplementary Fig. 3b) with enhanced cellular uptake and ability to inhibit KDM6A/B in cells following release of the free acid in the cytoplasm[36]. Finally, recognizing the importance of the GSK-J4's pyridyl-pyrimidine bidentate interaction with the catalytic metal led to the development of inactive control compound GSK-J5 (Supplementary Fig. 3b). No other KDM6 inhibitors are listed on the Chemical Probes Portal; thus GSK-J4 and its matched target-inactive GSK-J5 remain unique tools to interrogate KDM6A/B function in cells.

For GSK-J4 study, 153 publications met the eligibility criteria (Supplementary Fig. 5, Supplementary Table 6). GSK-J4 must be hydrolysed in the cytoplasm and hence its recommended in-cell concentration on the Chemical Probes Portal is up to 5 μM. 72 publications (47%) did not exceed this concentration in cell-based assays. The number of publications employing GSK-J4 within the 5 μM – 10 μM range (probably linked to GSK-J4's cellular $IC_{50}$ of 9 μM in macrophages[36], a value listed on the Structural Genomics Consortium's Chemical Probes website) was 52 (41%). Hence, 124 publications (81%) fully complied with the recommendations of the Chemical Probes Portal or Structural Genomics Consortium's Chemical Probes website. We found 10 publications (7%) that used GSK-J4 below and above 10 μM. However, only 16 publications (11%) employed GSK-J5 to validate cellular effects of GSK-J4 as a consequence of KDM6 inhibition (Fig. 2c).

## CREBBP/p300 chemical probe A-485

Lysine acetylation by histone acetyltransferases is another prevalent post-translational modification on histone tails and leads to the activation of gene transcription. Cyclic-AMP response element binding protein (CREB) binding protein (CREBBP, also known as CBP or KAT3A) and its paralog E1A binding protein p300 (also known as p300 or KAT3B), functioning as histone acetyltransferases and transcriptional co-factors, are indispensable for a multitude of cellular processes.

Given their high structural similarity and functional redundancy, they are often called CREBBP/p300 (or CBP/p300) and a significant focus has been placed on the development of CREBBP/p300 inhibitors[36,37].

Several inhibitors targeting either the catalytic acetyltransferase domain or the reader bromodomain of CREBBP/p300 have been endorsed as chemical probes (Supplementary Fig. 6). The chemical probe A-485 selectively targets CREBBP/p300 and its matched inactive analogue A-486 is 1000-fold less potent[38]. The more recently developed CPI-1612 engages CREBBP/p300 in cells at 50 nM, making it one of the most potent epigenetic chemical probes[39]. The remaining chemical probes target CREBBP/p300 bromodomains with varying selectivity over bromodomains of other proteins. For example, CPI-637 shows >700-fold selectivity for CREBBP/p300 over the BRD4 bromodomain 1, however it inhibits the bromodomain of BRD9[40]. The structurally related GNE-781 and GNE-049 are both suitable to interrogate the function of CREBBP/p300 without the complication of BRD inhibition[41]. Chemical probe PF-CBP1, its matched bromodomain-inactive dimethyl derivative ISOX-INACT[42], as well as structurally related SGC-CBP30[43] bind the CREBBP bromodomain with some degree of selectivity over BRD4. Similarly, I-CBP112[44] displays selectivity for bromodomains of CREBPP and p300 over the bromodomains of BRD4 (Supplementary Fig. 6).

In this cohort, 30 eligible publications (Supplementary Fig. 7) were reviewed for the concentration of A-485 and whether the inactive control A-486 and orthogonal inhibitors were employed (Supplementary Table 8). The recommended highest in-cell concentration for A-485 is 800 nM and we identified 8 publications (27%) that employed A-485 up to 1 μM. Another 8 publications (27%) partially complied as the concentration of A-485 was higher in some experiments. Of the 30 publications, 3 publications (10%) used the inactive control A-486 and 12 publications (40%) employed orthogonal inhibitors (Fig. 2d), some of which are endorsed chemical probes (e.g., CPI-637; Supplementary Table 8).

## Aurora chemical probes AMG900 and AZD1152

The three mammalian Aurora kinases (AURKA, AURKB, AURKC) are serine/threonine kinases with major roles in mitosis and meiosis[45]. While a large number of Aurora kinase inhibitors have been developed, only four inhibitors are endorsed as chemical probes in the Chemical Probes Portal (Supplementary Fig. 8a). MK-5108 inhibits AURKA with picomolar in vitro and nanomolar in-cell potency[46]. AZD1152 (barasertib) is a phosphate pro-drug that is rapidly cleaved to the active alcohol AZD1152-HQPA, which potently and selectively inhibits AURKB[47]. While AZD1152 is suitable for in vivo studies, for cell-based experiments AZD1152-HQPA is a preferable chemical probe, since cleavage of the phosphate group in AZD1152 might not occur in cells. AMG900 inhibits all three Aurora kinases isoforms with nanomolar potency, and also has nanomolar affinity for several other kinases[48]. Hence, it is recommended to use AMG900 together with orthogonal chemical probes, such as XMD-12 which is among the most kinase selective pan-Aurora inhibitors reported. Compared to AMG900, XMD-12 also targets all three AURK isoforms, but shows a better kinome-wide selectivity below 100 nM[49].

We identified 21 publications (Supplemenatry Fig. 9) that employed AMG900 in at least one figure panel (Supplementary Table 10). Of these publications, 14 (67%) did not exceed the recommended 100 nM in-cell concentration, 4 (19%) employed AMG900 below 100 nM in some experiments and 10 (48%) confirmed AMG900 results with additional Aurora inhibitors, such as orthogonal chemical probes AZD1152 and MK-5108 (Fig. 2e). Our search for the more popular chemical probe AZD1152 identified 128 publications (Supplementary Fig. 10) presenting data obtained from cells treated with AZD1152 (Supplementary Table 12). Of these, 55 publications (43%) did not exceed the recommended 100 nM in-cell concentration. An additional 33 publications (26%) partially complied with the 100 nM

recommendations, and 50 publications (39%) included in their experiments orthogonal Aurora inhibitors (Fig. 2f).

## mTOR targeting probe AZD2014

Mammalian target of rapamycin (mTOR) is a serine/threonine protein kinase. mTOR is part of two structurally and functionally distinct mTOR complexes 1 and 2 (mTORC1/2) which regulate cellular growth and metabolism either through direct phosphorylation of kinases (e.g.; ribosomal protein S6 kinase, Akt kinase) or indirectly through downstream signalling effectors (e.g.; eIF4E binding protein, MYC, HIF1α, SREBP1). Due to its involvement in numerous signalling pathways, mTOR is a master regulator of cell growth and metabolism[50,51].

Bearing the name of the target is the chemical probe rapamycin (also known as sirolimus, Supplementary Fig. 8b), a potent inhibitor of, primarily, mTORC1[52]. Poor solubility and pharmacokinetics led to the development of two water-soluble synthetic derivatives everolimus and temsirolimus. A phosphine oxide pro-drug version of rapamycin, ridaforolimus, was also developed (Supplementary Fig. 8b). These analogues of rapamycin, entitled rapalogs, are more suitable for in vivo studies but given their structural similarities, they cannot be considered as orthogonal chemical probes to rapamycin. One of the most well-characterised mTOR chemical probes is the pyridopyrimidine-based compound AZD2014[53]. Based on an earlier mTOR inhibitor AZD8055[54], AZD2014 mimics the selectivity and potency of this earlier inhibitor, but with improved solubility and metabolic stability[55]. The pyrazolopyrimidine eCF309 was discovered as part of a campaign to develop inhibitors of both mTORC1 and 2[56].

In the AZD2014 search, we identified 93 publications fulfilling the eligibility criteria (Supplementary Fig. 11), which were reviewed for the AZD2014's concentration and use of orthogonal mTOR inhibitors (Supplementary Table 14). We identified 71 complying and 9 partially complying publications (76 and 10%, respectively) that followed the recommended maximal in-cell concentration of 2 μM. Of 93 eligible publications, 56 (60%) used orthogonal mTOR inhibitors, predominantly rapamycin and/or rapalogs (Fig. 2g).

## CDK7 chemical probe THZ1

Cyclin-dependent kinase 7 (CDK7) is a master regulator of cell-cycle progression and gene transcription[57]. CDK7, together with cyclin H and MAT1, comprise the CDK-activating kinase (CAK) complex that phosphorylates CDK1/2/4, leading to their full activation and cell cycle progression. Moreover, the CAK is a component of the multi-protein complex that is essential for RNA polymerase II (Pol II)-mediated transcription. However, whether CDK7 is required for Pol II phosphorylation has been challenged by a study demonstrating a non-essential role for CDK7 in Pol II phosphorylation and global gene expression[58]. Given the multiple oncogenic and some controversial roles for CDK7, development of CDK7 inhibitors has gained momentum in the past decade[57].

The chemical probe THZ1 (Supplementary Fig. 12) inhibits CDK7 by covalently binding to a cysteine residue (C312) that lies outside the canonical ATP-binding site of the kinase and leads to decreased phosphorylation of CDK7 substrates. The non-covalent analogue, THZ1-R, lacking the acrylamide functionality responsible for the covalent bond with C132, is used as an inactive control compound for THZ1[59]. Of note, THZ1 inhibits also CDK12/13 via the same covalent mechanism of action, and thus THZ1 is not ideally selective (hence the low Global Score in the Probe Miner, Table 1). To separate CDK7 and CDK12/13 inhibition, the chemical probe THZ531 was developed (Supplementary Fig. 12). THZ531 is a derivative of THZ1 with the same phenylaminopyrimidine core scaffold, which inhibits CDK12/13 approximately 20 times more potently than CDK7[60]. Importantly, two target-inactive control compounds accompany THZ531. In the analogue THZ531R the electrophilic acrylamide is replaced with a propyl amide incapable of a covalent bond, and THZ532 is the enantiomer of

THZ531 (Supplementary Fig. 12). Both analogues are 50- to 100-fold less active on CDK12 and CDK13[60]. A CDK7-selective covalent inhibitor YKL-5-124 was later developed by combining the pyrrolidinopyrazole core from a previously unexplored CDK-targeting scaffold with the covalent warhead from THZ1. Removal of the acrylamide functionality, and replacement with an ethyl chain, gave the inactive counterpart, YKL-5-167 (Supplementary Fig. 12)[58].

In the THZ1 cohort, 134 eligible publications (Supplementary Fig. 13) were included in the analysis (Supplementary Table 16). THZ1 is recommended for cell-based experiment at the maximal concentration of 1 µM and we found 124 (93%) compliant and 7 (5%) partially compliant publications. However, only 9 (7%) publications validated data with the inactive control compound THZ1R. Given that THZ1 targets CDK7 and CDK12/13 with comparable potency, analysed publications focused on the role of CDK7 and/or CDK12/13. As such, orthogonal inhibitors were targeting either CDK7 or CDK12/13; in some publications pan-CDK inhibitors were employed. All publications using any CDK inhibitor were deemed as compliant and in total 42 publications (31%) included an additional CDK inhibitor (Fig. 2h).

### Limitations

Before discussing the results of our analysis, it is important to recognise its limitations. Although we performed three independent searches for each chemical probe, there may be more publications based on cell-based experiments using a given chemical probe, which were not included. This unintentional oversight could have an impact on the percentages of compliance. It is plausible that we have missed studies correctly using the given chemical probe, which would increase compliance percentage. Furthermore, despite carefully reviewing each publication, there is a possibility that a small fraction of what appears to be non-compliant publications may in fact be a correctly designed study. The experimental design uniquely depends on the hypothesis being tested and some publications might have intentionally and for a valid reason used, for example, a higher concentration of a given chemical probe. To mitigate this limitation as much as possible, publications in which the chemical probes were used within a range of concentrations, some of which exceeded the in-cell recommendations, were regarded as partially complying if at least one figure presented data with a chemical probe below the recommended maximal concentration. This, however, could have resulted in suboptimal experimental designs counted as compliant, and thus unrealistically increasing the compliance percentage. Given that the eligible publications were across all research fields, many beyond our expertise, we did not feel confident to assess whether the chemical probe's concentration in each study was justifiable. We also recognize that 662 publications do not represent all literature corpora and our conclusions are only estimates of the status quo.

## Discussion

Good news and bad news have emerged from our analysis. The good news is that high quality chemical probes, developed in response to the robustness crisis, have been increasingly employed in basic and preclinical research. In the past decades, a high percentage of novel drug targets could not be validated and this was partially compounded by using compounds that were inadequately characterised, had been later shown to be non-selective, flawed or containing pan-assay interference (PAINS) elements[61,62]. Initiatives such as the NIH Molecular Libraries Program and Structural Genomics Consortium, as well as the dedication of the chemical biology community yielded a significant compendium of high-quality chemical probes. This resulted in one noticeable improvement: we found only few recent studies[63–66] using Historical compounds such as PF-03814735 which inhibits 123 targets and GSK-J1 which is not cell-permeable; or molecules classifying as PAINS (e.g.; C646)[67]. Another good news is that within the cohort of 662 analysed publications, only 22% have reported using chemical probes above the recommended in-cell concentration (Table 2). Of note, however, the non-compliance with the in-cell concentration varied greatly, ranging from 2 to 70% (Table 2). Moreover, when assessing the compliance with in-cell concentrations, we have given numerous studies 'the benefit of the doubt' and considered them partially compliant when at least one experiment was performed within the recommended range.

The bad news is that on average 58% of relevant publications ($n = 509$) used only one chemical probe and did not validate findings with orthogonal inhibitors. Of further concern is the quality of orthogonal inhibitors in the remaining (42%) compliant publications. We noticed that many studies relied on structurally related chemical probes, less selective molecules and/or pan-inhibitors at high(er) concentrations. Probably the most unexpected finding was the lack of matched inactive control compounds in 92% of relevant publications ($n = 420$, Table 2). Finally, within the cohort of 267 eligible publications related to UNC1999, UNC0638, A-485 and THZ1, chemical probes which are accompanied with matched target-inactive control compounds and orthogonal inhibitors, only a worryingly low 4% of publications (11/267) complied with all three recommendations: i.e., the chemical probe was below the recommended in-cell concentration in most figures and orthogonal as well as inactive control compounds were included in the study.

The consequence of the suboptimal (i.e., chemical probes at high concentration) or missing (i.e., lack of inactive and orthogonal compounds) experimental design would be best described as unfinished projects. It is predicted that the median number of protein targets per chemical molecule is up to 329 and thus even the highest quality chemical probes are not expected to be selective at the proteome scale[68]. Rather, the selectivity of every chemical probe depends on the concentration used in experiments. Therefore, to minimise

**Table 2 | Summary of non-compliance with guidelines for the use of selected chemical probes**

| Target | Chemical probe | Inactive control compound | Number of publications (Jan 2023) | | | |
|---|---|---|---|---|---|---|
| | | | Analysed | Using higher than recommended in-cell concentration (%) | Not using inactive control (%) | Not using orthogonal inhibitors (%) |
| EZH2 | UNC1999 | UNC2400 | 49 | 17 (35%) | 43 (88%) | 26 (53%) |
| G9a/GLP | UNC0638 | UNC0737 | 54 | 38 (70%) | 53 (98%) | 31 (57%) |
| KDM6 | GSK-J4 | GSK-J5 | 153 | 19 (12%) | 137 (89%) | n/a |
| CBP/p300 | A-485 | A-486 | 30 | 14 (47%) | 27 (90%) | 18 (60%) |
| pan-AURK | AMG900 | n/a | 21 | 3 (14%) | n/a | 11 (52%) |
| AURKA | AZD1152 | n/a | 128 | 40 (31%) | n/a | 78 (61%) |
| mTOR | AZD2014 | n/a | 93 | 13 (14%) | n/a | 37 (40%) |
| CDK7 | THZ1 | THZ1-R | 134 | 3 (2%) | 125 (93%) | 92 (69%) |
| | TOTAL | | | 147/662 **22%** | 385/420 **92%** | 293/509 **58%** |

Source data are provided as a Source Data file.

engagement of off-targets it is critical to apply chemical probes at the lowest feasible concentration and demonstrate target modulation in cells at this concentration. Further, chemical probes are largely used to link the primary target of the probe to a phenotype, thus a phenotype observed under a narrow set of conditions (i.e., after a treatment with one chemical probe) must be validated with orthogonal chemical probes and structurally matched target-inactive compounds. The ideal inactive compound does not target the protein of interest but retains the activity against the off-targets. However, the majority of control compounds chemically deviate from the chemical probe by a single heavy atom and even this minimal change to the structure can diminish activity against 80% of the probe's targets[69]. Therefore, although the inactive compounds should be used in parallel with the chemical probes, inclusion of two orthogonal chemical probes is particularly important for the robustness of findings. Employing orthogonal and inactive compounds only marginally increases the size of the experiment (e.g., a few more plates/wells to be treated, one or two additional lanes on immunoblots), however the associated results significantly increase data reliability. These low effort-high impact control experiments will finish the project by confirming that the phenotype obtained with a given chemical probe is indeed due to the targeting of the protein of interest, rather than a technical issue with the assay and/or off-targets of the probe.

The unfinished projects indirectly but significantly impact future research. Researchers generally do not precisely repeat the experiments reported by others. From a practical standpoint, in cancer research for example, the heterogeneity of cultured cells makes it often impossible to repeat one's experiments under the same conditions. Instead, researchers enquire whether the conclusions made by others, such as targeting a given protein kills cancer cells, is pertinent to their disease models. However, hypotheses that are guided by unfinished projects might generate unexpected results. Furthermore, scientists often reach out to publications for experimental design and unknowingly follow a suboptimal approach to answer their hypothesis. As a possible impact, we analysed 14,896 citations that 662 publications included in our analysis received by January 2023 (Table 3). We found that 2583 citations (17% of all citations) are linked to publications using chemical probes at higher than recommended in-cell concentrations; 7744 citations (83% of all relevant citations) are linked to publications not employing inactive control compounds and 7089 citations (59% of all relevant citations) are referring to publications using only one chemical probe. Given this high number of citations linked to unfinished projects, it appears that suboptimal application of chemical probes is an ongoing issue.

## Future directions

The quality and quantity of chemical probes continue to rise, and we are on the way to achieving full human proteome coverage thanks to initiatives such as Target 2035 (https://www.target2035.net/), which aims to create, by year 2035, chemogenomic libraries: chemical and/or biological probes for the entire human proteome[70]. However, while the chemical probes are becoming increasingly accessible, their correct use needs to be promoted much more widely and effectively. The chemical probes resources (Table 4) are user-friendly and recommendations for cell-based experiments using chemical probes can be found in reviews[6–9,11,12]. Free expert-delivered webinars (Table 4) are another easy way to understand chemical probes and learn how to apply them correctly. It is, however, important that scientists relying on research with chemical probes are informed about these resources. Therefore, those skilled with chemical probes could routinely include in their research talks and conference presentations a slide outlining the importance of best practice as well as listing resources to chemical probes and guidelines. The Chemical Probes Portal has developed a useful and informative pack of slides for this purpose (Table 4). Those less skilled in chemical probes should be encouraged to seek advice from experts in the field.

More than one molecular probe (e.g.; siRNA) and demonstrating phenotype in at least two cancer cell lines has become a standard requirement in many cancer journals. We would like to propose 'the rule of two' for chemical probes as well: at least two chemical probes (either orthogonal target-engaging chemical probes, or a pair of a chemical probe and matched target-inactive compound) to be employed at reasonable concentration in every study. If inactive and orthogonal probes are available for the targeted protein, both must be included in the experiments. During manuscript review process, a close attention should be given to this rule, and it is also critical that every manuscript provides evidence of target engagement and/or modulation of downstream pathways by a chemical probe within the recommended concentration range. These experiments should be performed in a hypothesis-relevant cell-based models and the concentration of the chemical probe that relates to the target engagement/modulation should be then applied through the whole study. Of note, while our publication focuses on chemical probes, it is necessary that correctly performed experiments with chemical probes are further validated with molecular probes.

Finally, we provide a simplified flowchart for researchers as a guide to select and use chemical probes correctly, as well as five checklist items that should be addressed when reviewing manuscripts (Fig. 3). Good practice can be achieved by frequent conversations and collaborations between probe developers and users. Good practice could be supported by a dedicated chemical biology reviewer or editor in all journals publishing manuscripts using chemical probes. By practicing good practice, we will use research funds more effectively and hopefully deliver trustworthy scientific breakthroughs at a faster pace.

**Table 3 | Summary of citations for publications identified as non-compliant with recommendations**

| Probe | Publications | Number of citations (Jan 2023) | | | |
|---|---|---|---|---|---|
| | | For all | For publications with higher than recommended in-cell concentration (%) | For publications not using inactive compound (%) | For publications not using orthogonal inhibitors (%) |
| UNC1999 | 49 | 909 | 220 (24%) | 650 (72%) | 327 (36%) |
| UNC0638 | 54 | 1273 | 811 (64%) | 1175 (92%) | 784 (62%) |
| GSK-J4 | 153 | 2890 | 216 (7%) | 2268 (78%) | n/a |
| A-485 | 30 | 503 | 295 (59%) | 403 (80%) | 158 (31%) |
| AMG900 | 21 | 271 | 5 (2%) | n/a | 231 (85%) |
| AZD1152 | 128 | 3189 | 682 (21%) | n/a | 2254 (71%) |
| AZD2014 | 93 | 2076 | 278 (13%) | n/a | 763 (37%) |
| THZ-1 | 134 | 3785 | 76 (2%) | 3248 (86%) | 2572 (68%) |
| TOTAL | 662 | 14,896 | 2583 / 14,896 (**17%**) | 7744 / 9360 (**83%**) | 7089 / 12006 (**59%**) |

Source data are provided as a Source Data file.

**Table 4 | Links to selected resources, webinars and slides on chemical probes**

| Resource type | URL |
|---|---|
| *Expert-curated portals* | |
| Chemical Probes Portal | www.chemicalprobes.org |
| Structural Genomics Consortium | https://www.thesgc.org/chemical-probes |
| Nathanael Gray Laboratory | https://graylab.stanford.edu/probe-resources/ |
| *Data-driven computational portals* | |
| Probe Miner | https://probeminer.icr.ac.uk/#/ |
| Probes & Drugs Portal | https://www.probes-drugs.org/home/ |
| Small Molecule Suite | https://lsp.connect.hms.harvard.edu/smallmoleculesuite/ |
| *Expert-delivered webinars* | |
| Chemical probes as essential tools for biological discovery | https://www.youtube.com/watch?v=ZthORK6mSLI |
| Best practices for validating chemical probes | https://www.youtube.com/watch?v=rBGU8CKskTE |
| Best practices for validating chemical probes: Case Study 1 – MALT1 protease inhibitors | https://www.youtube.com/watch?v=LuEqOLtzJlY&list=PLSvEKj6f3OHFeOeel4VZyHaLSPhVYFmec&index=7 |
| Target validation using chemical probes | https://www.youtube.com/watch?v=NEeaeeU6-po |
| Chemical probes in disease modelling approaches | https://www.youtube.com/watch?v=h45NUCRqZM4 |
| Target 2035: Resource landscape for chemical probes | https://www.youtube.com/watch?v=vCASeF9LAJ8 |
| *Presentation slides for inclusion in talks* | |
| Link to the slides pack developed by the Chemical Probes Portal's team | https://www.chemicalprobes.org/information-centre#presentations |

## Methods

Through the whole study, we adhered to the Preferred Items for Systematic Reviews and Meta-Analysis (PRISMA 2020) guidelines[19].

### Protocol and registration

We could not preregister the study as it did not include any health-related outcome. Details on the conception of this study and the protocol followed is described herein in this publication. All data and notes associated with it are provided in the Supplementary Information, Supplementary Data and Source Data files.

### Database search

Searches were performed in 2022, with last searches to identify most recent articles completed between 16–20 January 2023. We used the PubMed database (https://pubmed.ncbi.nlm.nih.gov) of the United States National Library of Medicine at the National Institutes of Health and the SciFinder[n] portal (https://scifinder.cas.org/) of the Chemical Abstract Service. All searches were performed with no language or time restrictions. The bibliographic records retrieved from a "text search" in SciFinder[n] are sourced from CAplus and MEDLINE databases, whereas records retrieved in SciFinder[n] substance search originate from CAS REGISTRY[71]. The chemical probe names (*i.e.*, UNC1999, UNC0638, GSK-J4, A-485, AMG900, AZD1152, Barasertib, AZD2014, THZ1) were used as the query in the PubMed and SciFinder[n] keyword searches. The SciFinder[n] structure-based search was completed with the structure of each chemical probe drawn into the structure query box. The outputs of these searches are provided in the Source Data file. The three searches for each chemical probe were combined, patents removed by an automation tool and duplicates/triplicates were removed manually, yielding a list of unique articles for each chemical probe. These searches were completed by JS and JRB.

### Study selection

In the first steps of screening, we excluded articles not written in English (Supplementary Note 1 - Reason #1) and articles whose full text were not available via University of Sydney library access (Reason #2). In the next step, we retrieved available full-text and English written articles. We excluded articles describing the discovery of a given chemical probe and/or medicinal chemistry articles using the probe as

the lead compound (Reason #3); articles in which the chemical probe was used to generate chemical probe-resistant clones and/or articles lacking mechanistic experiments (Reason #4); articles containing only in vivo (animal/human tissue) data and/or clinical trials (Reason #5) and reviews, commentaries, editorials, letters to the editors, conference proceedings, pre-prints and similar (Reason #6). The screening process was completed independently by JS and JRB, full-text articles were checked for eligibility criteria (Supplementary Note 1) by JS and LM. Individual PRISMA flow charts for each chemical probe and reasons for exclusion are provided in the Supplementary Information.

### Eligibility criteria

We included all primary research articles (referred to as 'publications') that contained at least one figure panel presenting results obtained with a given chemical probe in a cell-based assay, irrespective of the topics and research fields (Supplementary Note 1).

### Data items and data collection

For each eligible publication (listed in Supplementary Tables 2, 4, 6, 8, 10, 12, 14, 16), JS extracted the following data: (i) concentration of a given chemical probe used in cell-based assays; (ii) inclusion of structurally matched target-inactive control compound (if available) and (iii) list of orthogonal inhibitors/chemical probes. These three items were defined based on previous recommendations[9,11]. In the analysis of the chemical probes' concentrations, we assessed cell-based assays addressing a mechanistic investigation, target engagement, delineation of a signalling pathway and/or identification of a phenotype following treatment with a given chemical probe. The concentrations employed in cell viability and clonogenic assays were not considered as a range of concentrations above the recommended maximum is needed to construct dose-response curves for calculations of cellular $IC_{50}/EC_{50}$ values. Similarly, Cellular Thermal Shift Assay (CETSA, a direct assay to assess target engagement in cells)[72] or toxicity assessments require higher concentrations of compounds and these concentration values were omitted in our analysis. When addressing the use of inactive control compounds (where available), we analysed (yes/no) whether the control compound was included: either shown in at least one figure panel or mentioned but data not shown. When addressing the use of orthogonal inhibitors, we listed all

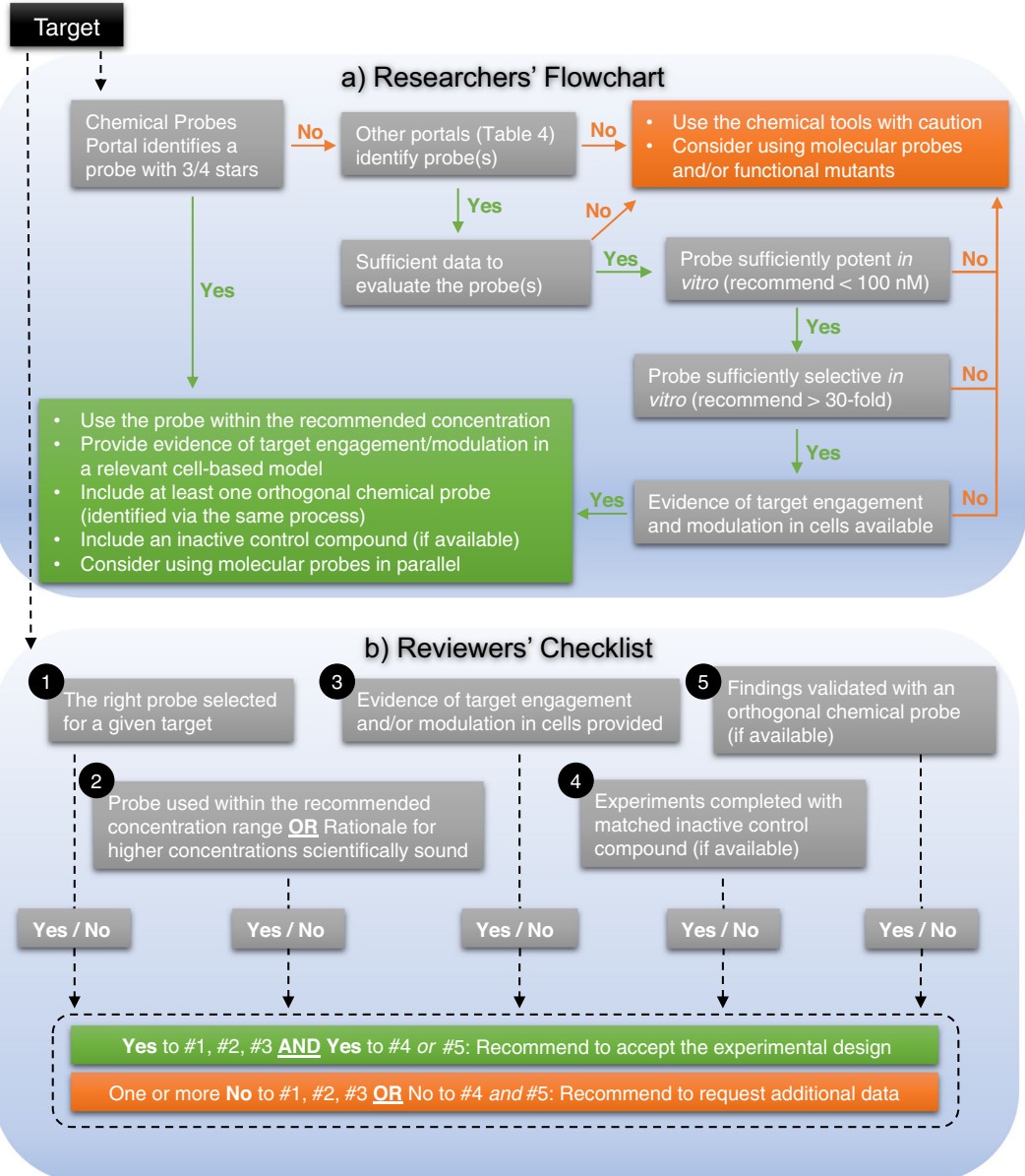

**Fig. 3 | Schematic summarising key steps for researchers and reviewers to correctly use chemical probes in cell-based research. a** For researchers, we recommend to start with the Chemical Probes Portal. If no probes are available on the Portal, other resources (listed in Table 4) can be used to identify a suitable probe. It is important to assess fitness factors of each probe and if the minimums are not met, we advise to use the chemical probe with caution and/or use molecular probes instead. **b** For reviewers, we recommend to accept the experimental design only if the right chemical probe (assessed by using the Flowchart) has been used within the recommended concentration range and evidence of target engagement/ modulation is demonstrated in a relevant cell-based assay (Yes to checklist items #1–3). To comply with 'the rule of two', at least one Yes is necessary to items #4 and #5.

chemical compounds that target the protein of interest and have been included in at least one figure panel. The number of citations were sourced from ScFinder[n]. There was a small number of publications not found in ScFinder[n] and for these publications citation numbers were sourced from the PubMed database.

## Synthesis of results and analysis

As the recommendations for chemical probes' maximum in-cell concentration, we used values listed on the Chemical Probes Portal (www.chemicalprobes.org) and/or on the Structural Genomics Consortium's Chemical Probes (https://www.thesgc.org/chemical-probes) websites. If the entire publication reported a given chemical probe below the recommended in-cell concentration, the publication was considered complying. We found numerous publications in which the chemical probe was applied at various concentrations, some of which were exceeding the recommended in-cell concentration. In this cohort, publications were considered partially complying if at least one experiment was performed with a concentration of a given chemical probe below the recommended maximum in-cell concentration. Finally, publications which used a given chemical probe at concentrations above the recommended values were considered non-complying. When addressing the use of matched inactive control compounds (where available), publications assessed with Yes were considered complying, and publications assessed with "No" considered non-complying. When assessing the use of orthogonal inhibitors, we considered every publication using at least one orthogonal inhibitor (regardless of their fitness factors and/or concentration) complying. We did not limit the compliance assessment to the application of endorsed orthogonal chemical probes. Publications not using any orthogonal chemical compounds were deemed non-

complying. All details are tabulated in the Supplementary Tables 2, 4, 6, 8, 10, 12, 14, 16; with compliant items highlighted in blue font. The percentages of compliance for each chemical probe were calculated as the sum of complying/partially-complying/non-complying publications over the number of included publications, the summary is presented in Fig. 2 prepared with GraphPad Prism 9 software. For the pooled analysis of data (Table 2), the percentages were calculated as a sum of all non-complying publications over the total number of all relevant eligible publications. Similarly, citations percentages (Table 3) were calculated as a sum of citations for all non-complying publications over the sum of all relevant citations (Supplementary Notes 2–9).

### Reporting summary

Further information on research design is available in the Nature Portfolio Reporting Summary linked to this article.

## Data availability

All data generated in this study are provided in this manuscript, Source Data and Supplementary Information files. Completed PRISMA checklists are available as Supplementary Data 1. Results of searches containing all considered records are available as Supplementary Data 2. Source data are provided with this paper.

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

## Acknowledgements

The authors thank Paul Workman for discussions, support and insightful comments during the preparation of this manuscript. We thank Albert Antolin for help with the Probe Miner database.

## Author contributions

J.S. performed literature searches, detailed analysis of all eligible publications and prepared Supplementary Tables 1–16 and individual PRISMA flowcharts (Supplementary Figs. 2, 4, 5, 7, 9, 10, 11, 13). J.R.B. performed literature searches, prepared Supplementary Figs. 1, 3, 6, 8, 12 and drafted chemistry sections on all chemical probes, with input from A.M. L.M. conceptualised the study, supervised the analysis of publications, performed data curation, prepared Tables 1–4 and Figs. 1–3. L.M. wrote the manuscript and all authors contributed with editing.

## Competing interests

The authors declare no competing interests.
