## [Peer Review File · Nature Communications]

Systematic literature review reveals suboptimal use of chemical probes in cell-based biomedical researchREVIEWER COMMENTS

Reviewer #1 (Remarks to the Author):

Sterling and colleagues present here an analysis of the use of chemical probes in cell-based studies. Focusing on eight probes with epigenetic and kinase targets, they evaluated papers published in mid-2022 for use 1) at the proper concentrations in cells, 2) in tandem with inactive analogs, and 3) alongside orthogonal inhibitors. They find that only 5% of papers use all three methods, with some encouraging data suggesting that more researchers are at least using the right concentrations in cells.

This study addresses an important topic that is often underappreciated. The proliferation of papers over the years using poor-quality probes has muddied the scientific literature, and efforts to develop more rigorous chemical probe criteria have helped raise awareness to this issue. However, if researchers are still not adhering to these recommendations, there will remain a problem in interpreting the literature. Hopefully this paper can provide heightened awareness, and even spawn additional studies in different scientific areas (e.g., beyond epigenetic and kinase targets).

The manuscript is very well-written and clear, and I found the analyses to be rigorous. I have only two suggestions for revision:

1. In each section, the number of articles found is reported, as well as the excluded articles. It would be helpful to refer back to the Methods for inclusion criteria.
2. In the Discussion, it was stated that "only a few recent articles using poor quality molecules" -- inclusion of that data would be helpful. Apologies if I just missed that.

Reviewer #2 (Remarks to the Author):

The manuscript represents a useful resource for the scientific community.

The utilization of appropriate chemical probes is critical to associate protein modulation to the intended pharmacology. The inability to selectively modulate single proteins using the correct concentration of probes, the usage of inactive controls, will have downstream consequences with million of dollars investments and no medical advances.

The publication is well written - maybe a little repetitive in some cases.

A few suggestions are highlighted in the below attached documents.

I suggest the authors to address the suggestions and I would then support publication.

Reviewer #3 (Remarks to the Author):

Sterling et al examined the use of a subset of popular chemical probes in cell-based research studies to determine if the compounds were employed in compliance with their known limitations and with suitable controls in place. The authors used established criteria for chemical probe use that have been previously published by chemical probe experts. Specifically, the authors undertook an analysis of 8 chemical probes that have been extensively used to advance chemical biology, drive therapeutic discovery, and serve as a springboard for additional studies in other labs. The results of these analyses show that user awareness and adherence to established principles of chemical probe use affects (a) the quality of studies in which probes are integrated and (b) the conclusions derived thereof from which new research and milestones may be derived. The authors raise an issue that surfaces repeatedly in manuscript reviews, in scientific group discussions and in online scientific blogs, and is a subject of multiple papers and reviews over the years on the topic of appropriate chemical probe use. The subject continues to be significant as papers continue to be published using poorly characterized probes, assay conditions that are suboptimal for the probe employed, or don't include proper controls to which the probe should be compared for proper benchmarking.

Important aspects of the article include (1) raising awareness of the issue to the scientific community to improve studies that include the use of chemical probes, and (2) collating and providing links to resources for investigators to review chemical probes for given protein targets, associated chemical probe data, and criteria on how chemical probes are appropriately employed in cell assays. Referencing of key reviews and past recommendations and guidelines for chemical probe use are suitable in this paper. For transparency and to help readers understand what articles are in/excluded in the analyses of each probe, the reviewer suggests that the authors do include these tables and information in the SI rather than only including them for review.

This is a highly relevant topic that affects multidisciplinary science. Execution of studies that employ probes incorrectly or without proper controls affects the primary studies as well as those that build from that work, and this amplifies the chances of error across work in multiple areas of investigation in the basic science and drug discovery continuum. The authors are correct to bring awareness to the issue on

a high impact platform; however, there are concerns with the methodology and statistics used to arrive at the conclusions reported in this manuscript. This makes publication of this manuscript in this journal premature at this time. Revisions that address the following may alter this view.

Some comments and recommendations are listed below:

1. The statistics and methodology used to formulate the conclusions should be more robust. The authors selected 8 probes for this analysis. It is not reasonable to expect that the authors would be able to assemble a complete picture of how big of an issue this is statistically given that the number of existing chemical probes and associated references is unmanageable. However, eight probes in of itself seems like a small representative number of probe compounds. Do the authors make their point with the selected probes? Yes. However, given the relatively few probes selected for the analysis, in concert with limitations of the search methods, the statistics are likely not accurate and thus, weaken the authors' arguments.

Specifically:

A. the authors note that their search for each probe was based on a keyword search in PubMed. A search based on the chemical name and a structural search in an orthogonal database such as SciFinder and PubChem (or others) are good complements to the search already performed since the chemical probe may not be listed in the paper by its common name or moniker. For instance, PubChem shows that UNC1999 is known by several other names outside of the UNC1999 and IUPAC designated name, so a structural search may be more complete. Minimally, it would be helpful to know if an orthogonal structure search changed the number of papers for each probe that would be considered in the evaluation, and if so, in what way are the statistics altered, if at all? Minimally, a more complete analysis around each of the 8 probes is warranted if the authors limit their analyses to only this group of compounds.

B. In the section, "Review of full text publications" the authors note that they excluded from consideration the papers with those assays based on cell viability or determination of IC50/EC50. An issue with this is that it is also common to see investigators using chemical probes well beyond the reported and known solubility limits of the chemical probe and then the team reports a selectivity index that is not meaningful. This skews interpretation and sends a ripple effect through the readership who repeat the error, not only with the chemical probe, but with newer and "improved" compounds which may not be that much better as they are also used beyond their solubility limits. Therefore, articles that have cell viability and EC50 determining assays are relevant to this analysis and should not be excluded.

C. The authors note that they did not include "3) Review or clinical trial articles, as well as commentaries, editorials, letters, and similar.." – excerpt from section entitled, "Database search and

selection of articles for analysis.” Perhaps the authors can clarify as the exclusion of “letters and similar” is vague. For example, ACS Medicinal Chemistry Letters or Bioorganic and Medicinal Chemistry Letters would be at least two resources in which letters appear that are likely to use chemical probes and would be relevant. If the authors meant that opinion-based pieces were excluded, and not primary research references that appear as Letters, then this should be simply clarified in the language.

2. It would be good to include some of the specific recommendations made by the authors in the abstract rather than leaving that as a statement that forward paths are proposed.

3. As the authors note, traction on this requires adoption by the investigators and a check at the review and journal editor level. This is not a requirement, but perhaps the authors might consider a graphic or flow chart to insert that will be easily visible to readers to check box that they have met the recommended chemical probe criteria for their assays. These types of visuals can help investigators and reviewers be more mindful of the elements they should be paying attention to and allows them to easily port that graphic into slides for teaching purposes and slide decks for scientific presentations. Moreover, it may be a visual that resonates with editors to redraft for inclusion in submission and review criteria.

Response to reviewers' comments

Reviewer #1

The manuscript is very well-written and clear, and I found the analyses to be rigorous. I have only two suggestions for revision:

1. In each section, the number of articles found is reported, as well as the excluded articles. It would be helpful to refer back to the Methods for inclusion criteria.

Details of inclusion and exclusion criteria are provided in the Methods section (page 13-15), briefly summarised in the Results on page 4. Details of excluded publications and reasons for exclusion are listed in the Supplementary Information.

2. In the Discussion, it was stated that "only a few recent articles using poor quality molecules" -- inclusion of that data would be helpful. Apologies if I just missed that.

References have been added on the page 11.

Reviewer #2 (comments copied from the pdf file provided by this reviewer)

The publication is well written - maybe a little repetitive in some cases.

We have removed repetition as much as possible without losing content and reduced the manuscript ~ 6,300 words.

A few suggestions are highlighted in the below attached documents.

Page 2: In my opinion "chemical probes" can be used in preclinical research and "drugs" should be used to identify compounds used in the clinic.

This has been amended on page 2.

Page 4: Can the authors justify why 188 publications were excluded?

Details of inclusion and exclusion criteria are provided in the Methods section (page 13-15), briefly summarised in the Results on page 4. Details of excluded publication and reason for exclusion are listed in the Supplementary Information

Page 6. Useful tool for the scientific community - summary of relevant chemical probes and how to best design negative/inactive probes. We appreciate this positive feedback.

Page 14: Explain rationale for exemplary chemical probes selected for the study.

Rationale for the selection of eight chemical probes used in this systematic review is provided on page 4.

Reviewer #3

For transparency and to help readers understand what articles are in/excluded in the analyses of each probe, the reviewer suggests that the authors do include these tables and information in the SI rather than only including them for review.

We agree and support to publish the Supplementary Information containing details of 1,132 publications.

1. The statistics and methodology used to formulate the conclusions should be more robust. The authors selected 8 probes for this analysis. It is not reasonable to expect that the authors would be able to assemble a complete picture of how big of an issue this is statistically given that the number of existing chemical probes and associated references is unmanageable. However, eight probes in of itself seems like a small representative number of probe compounds. Do the authors make their point with the selected probes? Yes. However, given the relatively few probes selected for the analysis, in concert with limitations of the search methods, the statistics are likely not accurate and thus, weaken the authors' arguments.

The rationale for the selection of with chemical probes has been added to the Results sections on page 4. We agree that eight chemical probes do not assemble the complete picture and provide this statement in the Limitation section on page 10. However, given that for eight chemical probes we already retrieved 1,131 unique records, of which 662 were analysed, this represents a significant amount of work. Given that this type of analyses cannot be automated, analysing literature associated with e.g., 50 chemical probes is not manageable. We believe that these initial analyses

could stimulate further work on additional chemical probes. For example, a scientist working in the field of EZH2 biology could analyse the use of all EZH2 probes to provide a better overview of the quality of papers related to EZH2.

A. the authors note that their search for each probe was based on a keyword search in PubMed. A search based on the chemical name and a structural search in an orthogonal database such as SciFinder and PubChem (or others) are good complements to the search already performed since the chemical probe may not be listed in the paper by its common name or moniker. For instance, PubChem shows that UNC1999 is known by several other names outside of the UNC1999 and IUPAC designated name, so a structural search may be more complete. Minimally, it would be helpful to know if an orthogonal structure search changed the number of papers for each probe that would be considered in the evaluation, and if so, in what way are the statistics altered, if at all? Minimally, a more complete analysis around each of the 8 probes is warranted if the authors limit their analyses to only this group of compounds.

We performed additional analyses as suggested. PubChem searches retrieved less records than PubMed searches for every chemical probe. Regarding other names for a given probe, for example: UNC1999 exists in PubChem as UNC 1999 and UNC-1999. Using these search terms in PubMed identified less publications than using the most common name UNC1999 as we present in the manuscript. However, SciFinder searches (both with the name and structure) retrieved more records for each chemical probe. These records were all included in our analyses, duplicates removed and eligible publications analysed. Details of all searches and literature analyses are provided in Supplementary Information (flow charts and Tables). In total, the revised version of our manuscript includes additional 215 publications included in the analyses. These additional publications had low impact on compliance percentages (~10% difference), as outlined in Table below.

Exemptions are changes to GSK-J4 compliance (highlighted), which however are due to the different in-cell maximum concentration used for the evaluation. In the original submission we considered compliant publications only those that used GSK-J4 below 5 μ M as recommended by the Chemical Probes Portal. However, given that the Structural Genomics Consortium recommends using GSK-J4 up to 10 μ M, in the revised analysis all publications containing GSK-J4 up to 10 μ M were considered as complying. This caused a significant jump in full compliance (from 51% to 81%) and in parallel a significant decrease in partial compliance (from 40% to 7%). These changes are not due to the additional papers being analysed. We used the same approach for UNC1999 (two different maximum in-cell recommendations).

Probe	Compliance in ORIGINAL submission	Compliance in REVISED submission	Difference
UNC1999	Conc Fully: 14/32 (44%) Conc Partially: 7/32 (22%) Inactive ctr: 4/32 (13%) Orthogonal cmpds: 14/32 (44%)	Conc Fully: 18/49 (37%) Conc Partially: 14/49 (29%) Inactive ctr: 6/49 (12%) Orthogonal cmpds: 23/49 (47%)	- 7% +7% -1% +3%
UNC0638	Conc Fully: 7/36 (19%) Conc Partially: 6/36 (17%) Inactive ctr: 1/36 (3%) Orthogonal cmpds: 13/36 (36%)	Conc Fully: 9/54 (17%) Conc Partially: 7/54 (13%) Inactive ctr: 1/54 (2%) Orthogonal cmpds: 23/54 (43%)	-2% -4% -1% +7%
GSK-J4	Conc Fully: 54/106 (51%) Conc Partially: 42/106 (40%) Inactive ctr: 13/106 (12%)	Conc Fully: 124/153 (81%) Conc Partially: 10/153 (7%) Inactive ctr: 16/153 (11%)	+30% -33% -1%
A-485	Conc Fully: 2/14 (14%) Conc Partially: 3/14 (21%) Inactive ctr: 2/14 (14%) Orthogonal cmpds: 6/14 (43%)	Conc Fully: 8/30 (27%) Conc Partially: 8/30 (27%) Inactive ctr: 3/30 (10%) Orthogonal cmpds: 12/30 (40%)	+13% +6% -4% -3%
AMG900	Conc Fully: 6/11 (55%) Conc Partially: 3/11 (27%) Orthogonal cmpds: 5/11 (45%)	Conc Fully: 14/21 (67%) Conc Partially: 4/21 (19%) Orthogonal cmpds: 10/21 (52%)	+12% -8% +7%
AZD1152	Conc Fully: 40/78 (51%)	Conc Fully: 55/128 (43%)	-8%

	Conc Partially: 19/78 (24%) Orthogonal cmpds: 20/78 (26%)	Conc Partially: 33/128 (26%) Orthogonal cmpds: 50/128 (39%)	+2% +13%
AZD2014	Conc Fully: 46/61 (75%) Conc Partially: 6/61 (10%) Orthogonal cmpds: 32/61 (52%)	Conc Fully: 71/93 (76%) Conc Partially: 9/93 (10%) Orthogonal cmpds: 56/93 (60%)	+1% ±0% +8%
THZ-1	Conc Fully: 103/109 (94%) Conc Partially: 5/109 (5%) Inactive ctr: 8/109 (7%) Orthogonal cmpds: 36/109 (33%)	Conc Fully: 124 /134 (93%) Conc Partially: 7/134 (5%) Inactive ctr: 9/134 (7%) Orthogonal cmpds: 42/134 (31%)	-1% ±0% ±0% -2%

B. In the section, “Review of full text publications” the authors note that they excluded from consideration the papers with those assays based on cell viability or determination of IC₅₀/EC₅₀. An issue with this is that it is also common to see investigators using chemical probes well beyond the reported and known solubility limits of the chemical probe and then the team reports a selectivity index that is not meaningful. This skews interpretation and sends a ripple effect through the readership who repeat the error, not only with the chemical probe, but with newer and “improved” compounds which may not be that much better as they are also used beyond their solubility limits. Therefore, articles that have cell viability and EC₅₀ determining assays are relevant to this analysis and should not be excluded.

We respectfully disagree with this point. To determine IC₅₀/EC₅₀ values, point dilutions of a given inhibitor/chemical probe are necessary to construct a dose-response curve. In the dilution row, some concentrations must be above the recommended limits to achieve ~100% inhibition. Solubility and selectivity index are important issues; however we feel these are not relevant to our analysis – we asked whether the probe was used at the recommended concentration in the mechanistic assays. The recommended in-cell concentrations account for the solubility and selectivity index. If we include concentration values used in the dose-response viability assays, the % compliance will be lower (*i.e.*; less papers complying) which we believe would be an unreliable estimation of compliance.

C. The authors note that they did not include “3) Review or clinical trial articles, as well as commentaries, editorials, letters, and similar..” – excerpt from section entitled, “Database search and selection of articles for analysis.” Perhaps the authors can clarify as the exclusion of “letters and similar” is vague. For example, ACS Medicinal Chemistry Letters or Bioorganic and Medicinal Chemistry Letters would be at least two resources in which letters appear that are likely to use chemical probes and would be relevant. If the authors meant that opinion-based pieces were excluded, and not primary research references that appear as Letters, then this should be simply clarified in the language.

The word “Letters” has been changed to “Letters to the editors” and details of exclusion criteria are listed in Methods, page 14.

2. It would be good to include some of the specific recommendations made by the authors in the abstract rather than leaving that as a statement that forward paths are proposed.

Given the word limit (150 words) for the Abstract and necessity to comply with PRISMA 2020 guidelines we were limited to fully address this recommendation. Yet, we included the “rule of two” recommendation into the Abstract which we believe is the most important step forward.

3. As the authors note, traction on this requires adoption by the investigators and a check at the review and journal editor level. This is not a requirement, but perhaps the authors might consider a graphic or flow chart to insert that will be easily visible to readers to check box that they have met the recommended chemical probe criteria for their assays. These types of visuals can help investigators and reviewers be more mindful of the elements they should be paying attention to and allows them to easily port that graphic into slides for teaching purposes and slide decks for scientific presentations. Moreover, it may be a visual that resonates with editors to redraft for inclusion in submission and review criteria.

We addressed this excellent suggestion and prepared a new Figure 8 which contains steps to select the best available chemical probe (applicable to researchers) and check list of items to consider by reviewers.

REVIEWERS' COMMENTS

Reviewer #1 (Remarks to the Author):

The authors have done an excellent job responding to the reviewers' comments and suggestions. I endorse publication - this is an important topic for the community, and I hope to see more studies in this area in the future.

Reviewer #2 (Remarks to the Author):

The authors have carefully addressed the reviewers suggestions.

I suggest move forward with publication.

Reviewer #3 (Remarks to the Author):

The authors have integrated changes and responded to the reviewer suggestions and comments. It is suitable for publication.